# Site Selection and Layout of Material Reserve Based on Emergency Demand Graduation under Large-Scale Earthquake

**Xinxin Yan** [1,*], **Hanping Hou** [1,*], **Jianliang Yang** [2] and **Jiaqi Fang** [1]

1   School of Economics and Management, Beijing Jiaotong University, Beijing 100000, China; 17113150@bjtu.edu.cn
2   School of Economics and Management, Beijing University of Chemical Technology, Beijing 100000, China; yangjianliang@mail.buct.edu.cn
*   Correspondence: 18120598@bjtu.edu.cn (X.Y.); hphou@bjtu.edu.cn (H.H.); Tel.: +86-1870-162-1697 (X.Y.); +86-1371-880-7962 (H.H.)

**Abstract:** Reasonable siting layout of reserve emergency supplies plays a critical role in rapid response and accurate rescue after disaster. People's life safety and health, as well as the psychological satisfaction brought by the government's excellent emergency rescue level, is an important guarantee for maintaining social stability and sustainable development. Based on the coverage model, considering demand graduation, this paper develops a bi-objective optimization model to determine the optimal location plan of graduated supplies by maximizing the rescue satisfaction and minimizing the number of warehouses. A heuristic multi-center clustering location algorithm is designed to solve the model. This model is applied to the prepositioning of emergency supplies in an earthquake affected area in Sichuan province, China to verify the effectiveness of the model and algorithm. Finally, the paper discusses the influence of demand graduation on the location of emergency supplies. The results show that reasonable location planning of different levels of supplies can effectively improve the rescue satisfaction.

**Keywords:** demand graduation; emergency management; reserve siting layout

## 1. Introduction

The occurrence of large-scale earthquakes has claimed countless lives and caused massive destruction to infrastructure in the last decade [1,2]. After the disaster, the chaos of post disaster emergency rescue can easily lead to short-term and hasty decisions, leading to casualties, waste of resources, economic losses, and adverse effects on the ability of long-term goal of social sustainable development [3]. In order to minimize the occurrence of these unwise decisions, we must actively carry out disaster preparedness to improve the efficiency of rescue. The location and layout of emergency materials is one of the most important strategic planning in the disaster preparedness stage. Reasonable siting layout of reserve emergency supplies plays a critical role in rapid response and accurate rescue after disaster. People's life safety and health, as well as the psychological satisfaction brought by the government's excellent emergency rescue level, is an important guarantee for maintaining social stability and sustainable development.

According to the report of the international disaster (EM-DAT) in March 2019, about 10.6% of the deaths caused by natural disasters in 2018 were caused by earthquakes. After the earthquake, in addition to a small number of victims died immediately from the collision of hard objects formed by the collapse of buildings, resulting in the rupture of head or organs and other key organs, most of the injured were buried under the ruins waiting for rescue. The timely supply of medical drugs, water, bandages and other critical relief items is the crux to improve the survival rate of the victims [4]. Reasonable location and layout of emergency materials in disaster preparedness planning can effectively improve the speed of post disaster material distribution.

However, over the past experience, the rescue time has been delayed due to the untimely supply of urgent relief materials over times. On 25 April 2015, a strong earthquake occurred in Nepal, resulting in more than 8000 deaths, many injuries, destruction of medical facilities and a large number of houses. There is an urgent need to distribute key drugs to hospitals and other health facilities in an affected area [5]. On 12 November 2017, the Iran–Iraq border earthquake killed 620 people. During the response phase of the Kermanshah earthquake, Iran's food and Drug Administration (IFDA) faced significant delay in providing 28 kinds of drugs to meet the needs of survivors, including antibiotics, anti-diabetes, anti-bacteria, etc. [6]. Therefore, for urgent materials such as medicine and food, the key to reserve location before disaster is to ensure the response speed after disaster.

In addition, another example was the COVID-19 outbreak in early 2020 in China; according to the statement of the official platform of the Red Cross Society of China, a large number of emergency supplies from various organizations and groups poured into the distribution center at the beginning of the rescue, which caused great pressure on the distribution of materials. However, the emergency response of mask, protective clothing, alcohol, and other crucial materials is still slow. Many medical staff and patients are not protected, diagnosed, isolated, and treated in time, leading to death and the spread of the epidemic. The incident reflects the disorder of emergency relief materials distribution. There are similar problems in earthquake rescue. Through summing up the experience from this failure case, we found that after a large-scale disaster, large number of emergency materials are needed. For materials with low demand urgency, too fast response will disturb the rescue order and occupy limited resources. Some scholars also have realized this problem and they believe that priority can be set to deliver critical materials with high urgency to improve rescue efficiency. They assume that materials with different urgency are placed in the same reserve point. The urgent materials should be distributed first, and then the less urgent materials should be distributed next. This method has a disadvantage, when the location of the reserve point is far away, even if priority is given to the distribution of urgent materials, the response speed of urgent materials will still be very slow. On the basis of consulting a large number of literatures, many scholars have studied the material scheduling problems with different demand urgency and the material allocation problems considering the priority of disaster affected areas. However, The location and layout of emergency materials considering demand graduation is still a blank research field. Moreover, few people make an analyze on the impact factors of demand urgency and provide a detailed method to quantify the urgency level.

Remainder of the paper is organized as follows. The relevant literature is briefly reviewed in Section 2. In Section 3, the definition of the problem and the assumption of the model are described. Moreover, the mathematics model and the heuristic multi-center clustering location algorithm are presented. In Section 4, the description of case study and computational solution are presented. Discussion in regards to the influence of demand graduation on location and layout of emergency supplies is presents successively. In Section 5, concluding remarks and future research direction are provided.

## 2. Literature Review

For response speed and satisfaction rate is limited by the location and layout of emergency supplies [7], a considerable literature has grown up around the theme of location planning of emergency supplies. Horner and Downs proposed a warehouse location model to reserve relief items for the affected area in order to minimize the distribution cost of reliefs [8]. Galindo and Batta developed a model for pre siting emergency supplies for hurricane disasters, taking into account the possible damage or destruction of pre deployed supplies [9]. Evidence suggests that optimizing the location of emergency supplies can effectively increase the demand satisfaction rate of post disaster rescue and improve the reliability of emergency material support [10,11]. Several attempts have been made to prove that reasonable location planning of emergency supplies plays an essential role in timely distribution of emergency materials and shortening response time [12–14].

A number of literatures have been focused on the location and layout of emergency materials under uncertain demand [15–18], most of which made rational interval estimation for uncertain demand through fuzzy number [19], robust optimization [20], interval grey number [21] and stochastic scenario planning [22]. Gökalp studies the design of emergency logistics network in the disaster preparedness stage. The network design takes into account the fairness and sufficiency of services. Through the robustness of the model, any feasible solution can meet the needs of all scenarios. In this way, the model does not rely on any parameters that hard to estimate [23]. Duran designed an emergency material reserve location model, which determined a group of typical demand cases, and found the supply network configuration that minimized the average response time of all demand cases [24]. In fact, the key point of these works is transforming the uncertainty into certainty by parameter estimation and then study the location optimization problem. This method is applicable to the situation that the information of the disaster situation is not complete in the initial stage of an unknown disaster and cannot fully consider the urgency of the material needs of the victims.

Some scholars have realized that at different stages after the disaster, the urgency degree of the residents' demand for materials is different. Timely and accurate delivery of the materials most needed by the victims is the key to improve the rescue satisfaction [25,26]. How to develop the material graduation technique to scientifically determine the urgency level of materials has become a subject of extensive research [27,28]. Nevertheless, the majority of past research on demand graduation were applied to material storage [29] and distribution [30] et al. Prepositioning of emergency supplies considering demand graduation had been a largely under explored domain. The mainstream mathematical model of emergency supplies location originates from P-median and coverage model whose optimization objectives are maximum weighted distance and coverage or minimum facility quantity and cost [31,32]. Marcelin et al. proposed a p-median location model framework collaborating with a GIS for providing people with hurricane relief materials that aimed to minimize the total demand weighted travel costs [33]. Xue takes the financial efficiency and demand coverage rate as the key indicators to evaluate the emergency logistics management. Considering that the needs of disaster areas may be met by multiple organizations and facilities, a cooperative coverage model of emergency material distribution center considering budget constraints is established [34]. However, these models do not take the time effect value of rescue into account. In other words, rapid response to materials with low degree of urgency is not the most imminent. All levels of emergency supply should be accurately deployed according to the demand priorities so as to make the best use of the supplies and meet the urgent needs of the people.

This paper proposes a model for the reserve siting layout of emergency supplies in earthquake disaster to determine the minimum number of warehouses and optimization location of supplies so as to improve the response speed of post disaster emergency material rescue and the satisfaction of victims. The ultimate goal of the model is maximizing the rescue satisfaction and demand graduation is considered in the model construction. Moreover, the time effect value function under different demand levels is introduced to express the rescue satisfaction. The model is used to the location and layout planning of emergency materials in an earthquake prone area in Sichuan Province, China to verify the effectiveness of the model. Finally, the influence of demand graduation on the location and layout decision of emergency supplies is discussed.

## 3. Materials and Methods

### 3.1. Problem Description

After the occurrence of earthquake disaster, the emergency response law stipulates that the county-level emergency materials should be dispatched to the affected area as soon as possible. In order to deal with sudden disasters, the location and layout of county-level emergency materials within its jurisdiction should be planned. It is assumed that there are N counties in the region as demand points and they are also the alternative

emergency supplies prepositioning points. After the earthquake, the victims need V kinds of emergency materials.

According to the different urgency degree of the needs of the victims, V kinds of materials can be divided into l level. The distribution distance, delivery time, and the maximum rescue radius between the demand point and the alternative prepositioning point are known. The demand for various materials at the demand point can be calculated by the residential population, the proportion of disaster affected population, and the demand of various emergency materials per unit population. The prepositioning of emergency materials reserve is shown in Figure 1.

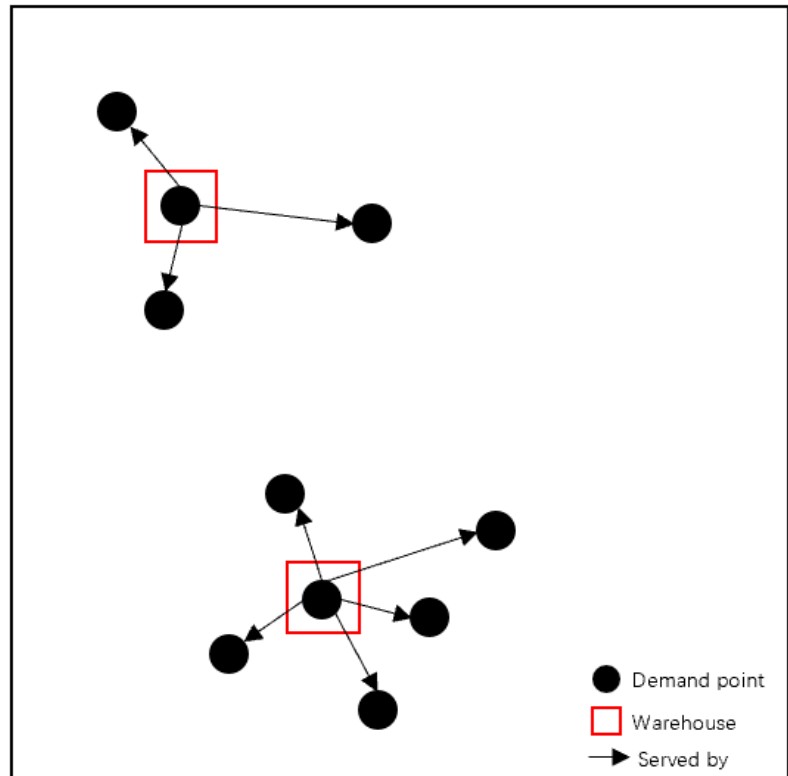

**Figure 1.** Prepositioning of emergency materials reserve.

The problems to be solved in the model are: reasonably divide the demand levels of various emergency supplies and locate different graded supplies at corresponding positions from the candidate points to make the best use of the time effect value of emergency materials. Before introducing the model, following assumptions are stated.

- For each demand point in the planning area, the probability and impact of disasters are the same.
- The emergency supplies are divided into three demand grades: extra urgent grade, urgent grade and general grade.
- Each demand point is rescued by a material reserve point.
- The transportation route is determined and the response time between any demand point and alternative reserve point is fixed.

### 3.2. Proposed Mathematical Model

3.2.1. Parameter Description

Before introducing the model, the parameters and variables in the model are described as follows.

| | |
|---|---|
| $I$ : | Set of demand points, $I = \{i \in I | i = 1, 2, \ldots, n\}$ |
| $J$ : | Set of reserve points, $J = \{j \in J | j = 1, 2, \ldots, n\}$ |
| $C$ : | Set of emergency materials, $C = \{c \in C | c = 1, 2, \ldots, v\}$ |
| $U$ : | Set of emergency grades, $U = \{u \in U | u = 1, 2, 3\}$ |
| $K$ : | Set of influencing factors of demand urgency, $K = \{k \in K | k = 1, 2, \ldots, h\}$ |
| $R_u$ : | Urgency interval fuzzy number of demand grade $u$ |
| $SR_u$ : | The lower limit of urgency interval fuzzy number for demand grade $u$ |
| $ER_u$ : | The upper limit of urgency interval fuzzy number for demand grade $u$ |
| $Grade_c$ : | Demand grade of material $c$, $Grade_c \in U$ |
| $G_{kc}$ : | Single factor evaluation value of influencing factor $k$ of emergency materials $c$ |
| $cor_{cku}$ : | The membership degree of material $c$ rated as grade $u$ on the influencing factor $k$, $cor_{cku} \in (0, 1]$ |
| $cor_{cu}$ : | The membership degree of material $c$ rated as grade $u$ on all factors, $cor_{cu} \in (0, 1]$ |
| $w_k$ : | Evaluation weight of influencing factor $k$ |
| $d_{ij}$ : | Distance from demand point $i$ to reserve point $j$ |
| $t_{ij}$ : | Response time of materials from demand point $i$ to reserve point $j$ |
| $p_i$ : | Residential population at demand point $i$ |
| $\varphi_i$ : | Proportion of affected population at demand point $i$ |
| $\gamma_c$ : | Demand of various emergency materials per unit population |
| $D_{iu}$ : | The demand quantity of $u$ grade materials at demand point $i$ |
| $D_{ic}$ : | Demand volume of material $c$ at point $i$ |
| $q_{iju}$ : | The supply of $u$ grade demand from reserve point $j$ to demand point $i$ |
| $q_{ijc}$ : | The supply of material $c$ from reserve point $j$ to demand point $i$ |
| $R_{max}$ : | Maximum rescue radius of reserve point |
| $S_u(t)$ : | The effect value function of $u$ grade demand |
| $x_j$ : | Select to set the reserve point as 1 at point $j$, otherwise it is 0 |
| $z_{ij}$ : | Demand point $i$ is served by reserve point $j$ |
| $y_{cu}$ : | Emergency material C belongs to u grade material, which is 1, otherwise it is 0 |
| $\theta$ : | Number of siting points deployed |

### 3.2.2. Reserve Siting Layout Model of Emergency Material Reserve Based on Demand Graduation

After the disaster, the victims have a large demand for all kinds of emergency materials. However, the urgent need of materials is different, the dispatching of all kinds of emergency materials should be divided into priority. Therefore, in the stage of disaster preparedness, all kinds of emergency materials should be graded scientifically and reasonably. Considering the time utility value of emergency materials, three factors that affect the urgency of emergency materials demand are determined, namely, the degree of life damage caused by insufficient supply, the degree of shortage of emergency materials and the degree of irreplaceable. On this basis, considering the volatility and variability of emergency material demand, they will have an impact on the demand urgency. The higher the volatility and the shorter the shelf life, the higher the sensitivity to time, and thus the higher the degree of demand urgency. Therefore, the emergency supplies grading index system based on the demand urgency is established by adding two influencing factors of demand periodicity and shelf life, as shown in Figure 2.

The demand level of all kinds of emergency materials is determined by Formulas (1)–(3) in the model. For any kind of emergency materials $c(c = 1, 2, \ldots, v)$, the evaluation value of five influencing factors of demand urgency $(G_{kc})$ is given through expert discussion. Formula (1) calculates the single factor membership degree $(cor_{cku})$ that emergency supplies $c$ can be rated as $u(u = 1, 2, 3)$ level on the influencing factor $k$. Formula (2) calculates the total factor membership degree $(cor_{cu})$ of emergency materials c which can be rated as $u$ level. Equation (3) is used to judge the demand grade of emergency materials c, which belongs to the level with the largest membership degree of all factors.

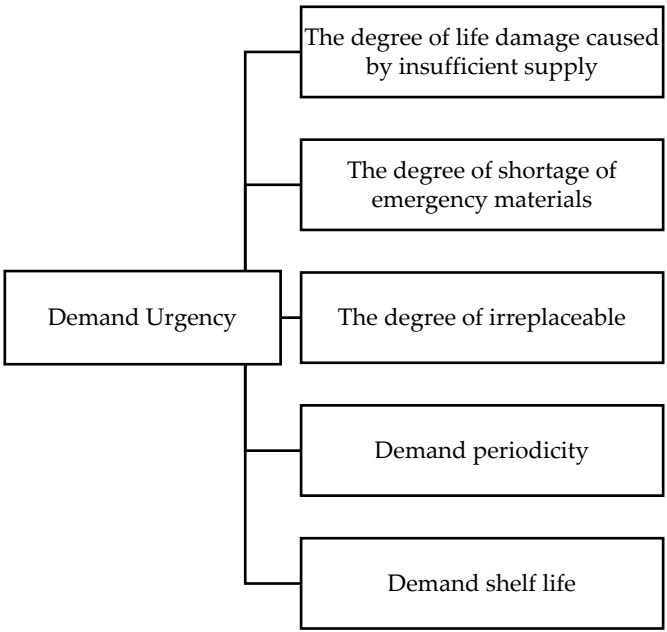

**Figure 2.** Graduation index system of emergency materials based on demand urgency.

$$cor_{cku} = e^{-|G_{kc} - \frac{1}{2}(SR_{ku} + E_{ku})|} \; \forall c \in C, \, k \in K, \, u \in U \tag{1}$$

$$cor_{cu} = \sum_{k=1}^{h} w_k cor_{cku} \; \forall c \in C, \, u \in U \tag{2}$$

$$Grade_c = u_0, \; if \; cor_{cu_0} = \max(cor_{cu}) \; \forall u \in U \tag{3}$$

After determining the level of emergency materials, the demand of each demand point for each level of emergency materials is calculated by (4) and (5).

$$D_{ic} = p_i \varphi_i \gamma_c \; \forall i \in I, \, c \in C \tag{4}$$

$$D_{iu} = \sum_{c=1}^{v} D_{ic} y_{cu} \; \forall i \in I, \, u \in U \tag{5}$$

Due to the different degree of urgency of demand, the satisfaction of victims is different when different grades of materials are delivered to the demand point in different rescue stages. In this paper, the time effect value function $S_u(t)$ is introduced to describe the rescue quality and the time effect value represents the satisfaction value of victims when the needs are met at different time. Table 1 shows the time effect value function of the three levels of demand. The trend of satisfaction of extra urgency and urgency materials is similar: during stage $0 - T_1$, the satisfaction value reaches to the peak at 1, then the satisfaction degree decrease gradually with the time delay between $T_1 - T_3$ and decline rate of extra urgency demand is faster than urgency demand. After $T_3$, it declines to 0. For general materials, at the first stage of $0 - T_1$, the satisfaction degree was 0.2. At the stage of $T_1 - T_2$, the satisfaction increases and reaches the peak at $T_2$. Between $T_2 - T_3$, the satisfaction falls and returns to 0.2 after $T_3$. As shown in Figure 3.

**Table 1.** Time effect value function.

|  | $S_u(t)$ | $t$ |
|---|---|---|
| $u = 1$ | 1 | $0 < t \leq T_1$ |
|  | $\left(\frac{t-T_3}{T_3-T_1}\right)^2$ | $T_1 < t \leq T_3$ |
|  | 0 | $t > T_3$ |
| $u = 2$ | 1 | $0 < t \leq T_1$ |
|  | $-\left(\frac{t-T_1}{T_2}\right)^2 + 1$ | $T_1 < t \leq T_3$ |
|  | 0 | $t > T_3$ |
| $u = 3$ | 0.2 | $0 < t \leq T_1$ |
|  | $-\frac{0.8}{T_1^2}(t - T_2)^2 + 1$ | $T_1 < t \leq T_3$ |
|  | 0.2 | $t > T_3$ |

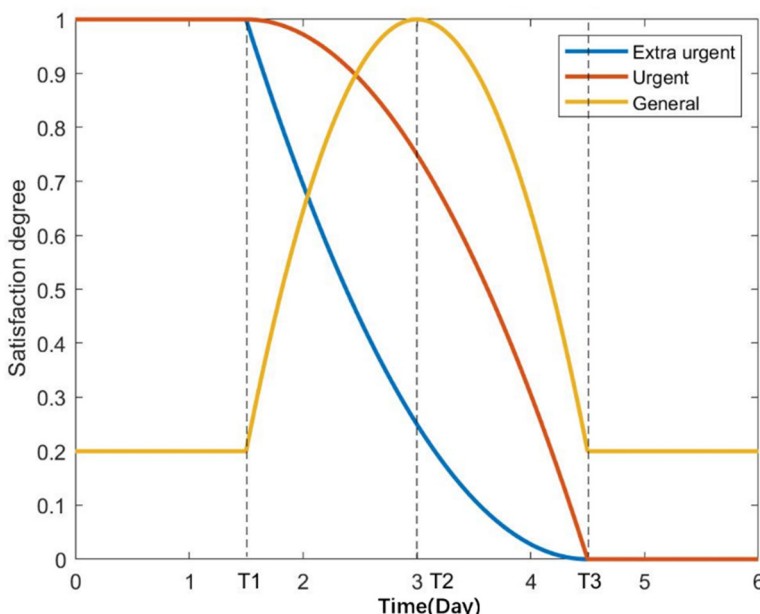

**Figure 3.** Time effect value under different demand levels.

The efficiency and quality of material rescue should be guaranteed first in the and layout of material reserve. Therefore, the maximum timeliness efficiency of emergency rescue is the primary goal of the model, as shown in Formula (6).

$$F_1 = max \sum_{u=1}^{l} \sum_{i=1}^{n} \sum_{j=1}^{\theta} S_u\left(t_{ij}\right) \tag{6}$$

It is well known that the quality of rescue can be improved by increasing the number of reserve points, but unlimited deployment of reserve points will cause waste of social resources and bring great pressure on government financial expenditure. Therefore, it is necessary to strike a balance between the quality of rescue and the number of reserve points. The secondary goal of the model is to minimize the total number of reserve points deployed. See Formula (7).

$$F_2 = min \sum_{j=1}^{n} x_j = \theta \tag{7}$$

Constraint (8) ensures that only the opened reserve point can provide rescue service for the demand point.

$$z_{ij} \leq x_j \ \forall i \in I, j \in J \tag{8}$$

Constraint (9) means that in the planning stage, only one reserve point is assigned to each demand point.

$$\sum_{j=1}^{m} z_{ij} = 1 \ \forall i \in I \tag{9}$$

Considering the emergency of the disaster, materials should be dispatched from the reserve point to the demand point in relatively short time. If the distance between the storage point and the demand point is far, and the material transportation time is too long, the best rescue time will be missed and disaster situation will be aggravated. Equation (10) constrains the reserve point to provide service only for the demand point within its rescue radius.

$$d_{ij}z_{ij} \le R_{max} \ \forall i \in I, j \in J \tag{10}$$

In order to avoid material waste, unnecessary storage and management costs are generated by stacking at the demand point. Therefore, constraint (11) is used to limit the quantity of materials provided by the reserve point to the demand point.

$$\sum_{j=1}^{m} q_{iju} \le D_{iu} \ \forall i \in I, u \epsilon U \tag{11}$$

$x_j$, $z_{ij}$, $y_{cu}$ in Formulas (12)–(14) are 0–1 decision variable. $q$ in Formular (15) is a nonnegative variable, and $\theta$ in Equation (16) is a positive integer.

$$x_j \in (0,1) \ \forall j \in J \tag{12}$$

$$z_{ij} \in (0,1) \ \forall i \in I, \ j \in J \tag{13}$$

$$y_{cu} \in (0,1) \ \forall c \in C, \ u \in U \tag{14}$$

$$q_{iju} \ge 0 \ \forall i \in I, j \in J, u \in U \tag{15}$$

$$\theta \in N+ \tag{16}$$

### 3.2.3. The Heuristic Multi-Center Clustering Location Algorithm

In the above problems, the number and location of reserve points are uncertain, so it is difficult to determine the optimal solution by accurate algorithm. Heuristic algorithm has advantages in solving such complex problems, so this paper uses heuristic algorithm to solve the model. The location problem can be regarded as the partition of the elements of a set to meet the distribution demand, which is similar to clustering analysis. Cluster analysis is an analysis method to classify things according to their similarity and find the characteristics of different categories of things [35]. The main research content is how to measure similarity and how to construct clustering method [36]. There are three basic methods of cluster analysis: system cluster method, decomposition method and dynamic method [37]. The basic idea of decomposition method is to treat all the demand points as one category, and then divide them into two or three categories according to the satisfaction of similarity until each requirement point can be grouped into an appropriate class.

Based on this idea, a heuristic multi center clustering location algorithm is proposed to solve the problem. The basic idea of the algorithm is: firstly, all the demand points are classified into one class, and then they are divided into two categories and three categories according to the constraint of the maximum rescue distance Until all requirement points are grouped into the appropriate class. The algorithm takes less time to solve the minimum number of reserve points to be deployed, so it is suitable for the problems in this paper.

In this paper, an optimization model with the main objective of "maximum time effect value" and "minimum number of reserve points" as the secondary objective is established. Since the optimization result of the secondary objective is the operation parameter of the primary objective, see Equations (6) and (7), the double objective cannot be solved

independently, and double nesting and simultaneous iteration are required. The detailed solution steps of the adaptive model are as follows:

(1)　Parameter definition and initialization.
(2)　Determine the demand urgency level of emergency materials.
(3)　Calculate the demand of each level of emergency materials at the demand point.
(4)　All demand points are classified into one class, and the value of secondary objective function is 1.
(5)　The optimal solution is the cluster center, which is the emergency material reserve point.
(6)　If the distances from all demand points to the cluster center satisfy the maximum rescue radius constraint, turn to step (9). Otherwise, turn to step (7).
(7)　All the demand points that do not meet the maximum rescue radius constraint are temporarily merged into a class, and the cluster center of the class is the optimal solution of the primary objective function; the secondary objective value is added by one.
(8)　Step (5) to step (7) are iterated until all demand points are assigned to the cluster centers satisfying the maximum rescue radius constraint.
(9)　Determine the service object and material configuration of each level. In order to satisfy the maximum distance constraint, the demand points belong to the nearest cluster center.
(10)　If demand points are assigned to the cluster centers satisfying the maximum rescue radius constraint, terminate the algorithm.

The algorithm flow chart is shown in Figure 4.

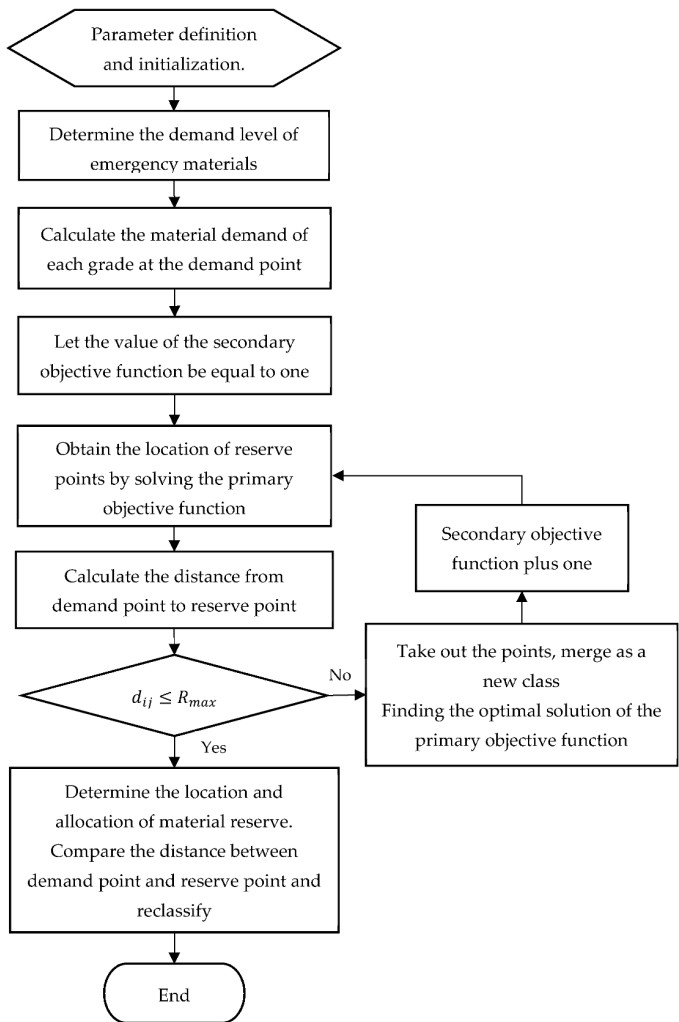

**Figure 4.** Flow chart of multi center heuristic clustering algorithm.

## 4. Results

### 4.1. Description of Case

In this case, 18 counties in Sichuan Province located in the Himalayan seismic belt are selected. The longitude and latitude coordinates and the local population are shown in Table 2. The number of people affected is generally 1/200 of the local population. After the earthquake, there are a large demand for six kinds of emergency supplies, such as food, bottled water, medical drugs, quilts, tents and shovels. The demand of six kinds of emergency materials for the affected population is 1, 1, 0.1, 0.25, 0.25, and 0.05, respectively. Each reserve point has six kinds of emergency materials, and the maximum rescue radius is 100 km. According to the grading index system of emergency materials proposed in 3.2, the weight values of the five influencing factors of demand urgency are 0.35, 0.12, 0.11, 0.24, and 0.18. The interval fuzzy numbers of three demand levels (extra urgent, urgent, and general) are [6, 9], [3, 6], [0, 3]. Table 3 shows the single factor evaluation matrix of the demand urgency of six categories of emergency materials by the emergency materials management expert group. Emergency rescue is divided into four stages, the first stage ends with node $T_1 = 1.5$ days, the end of the second stage node $T_2 = 3$ days, the third stage ends with node $T_3 = 4.5$ days.

**Table 2.** Longitude and latitude coordinates and local population information of 18 counties.

| No. | County | Lon. (°) | Lat. (°) | Population (Ten Thousand) | No. | County | Lon. (°) | Lat. (°) | Population (Ten Thousand) |
|---|---|---|---|---|---|---|---|---|---|
| 1 | Wei Yuan | 104.82 | 29.58 | 78.13 | 10 | Kang Ding | 101.75 | 30.20 | 13.58 |
| 2 | Gong Xia | 104.78 | 28.40 | 37.50 | 11 | Yue Xi | 102.74 | 28.38 | 37.30 |
| 3 | Chang Ning | 104.94 | 28.40 | 46.72 | 12 | De Qing | 99.33 | 28.20 | 6.00 |
| 4 | Xing Wen | 104.95 | 28.24 | 38.80 | 13 | Lu Shan | 103.00 | 30.34 | 11.95 |
| 5 | Xi Chang | 102.09 | 27.62 | 66.84 | 14 | Bai Yu | 99.40 | 30.95 | 5.36 |
| 6 | Qing Chuan | 105.05 | 32.25 | 21.28 | 15 | Lu Huo | 100.80 | 31.28 | 4.59 |
| 7 | Jiu Zhaigou | 103.82 | 33.20 | 8.15 | 16 | Jian Ge | 105.30 | 32.60 | 65.77 |
| 8 | Li Tang | 99.61 | 30.11 | 7.31 | 17 | Wen Chuan | 103.49 | 31.17 | 10.25 |
| 9 | Jin Kouhe | 103.20 | 29.30 | 5.27 | 18 | Dao Fu | 101.45 | 30.60 | 5.52 |

Data source: China seismological data center, Sichuan Provincial Bureau of Statistics.

**Table 3.** Single factor evaluation matrix of demand urgency of six types of emergency materials.

| Emergency Materials | The Degree of Life Damage Caused by Insufficient Supply | The Degree of Shortage of Emergency Materials | The Degree of Irreplaceable | Demand Periodicity | Demand Shelf Life |
|---|---|---|---|---|---|
| Food | 8.5 | 6.1 | 7.5 | 5.9 | 9.0 |
| Bottled Water | 8.7 | 5.8 | 7.4 | 6.2 | 8.6 |
| Medical Drugs | 9.0 | 8.8 | 7.5 | 6.4 | 6.1 |
| Quilt | 4.5 | 5.8 | 3.5 | 5.5 | 2.5 |
| Tent | 5.6 | 5.8 | 5.8 | 5.8 | 2.5 |
| Shovel | 2.4 | 2.5 | 1.5 | 1.5 | 1.0 |

### 4.2. Computational Results

(1) Demand grading results

Using MATLAB (R2017b) to solve the example, the running time is 4.74 s. Table 4 shows the membership degree values of all factors for each demand level of six types of emergency materials. According to Formula (3), emergency supplies should belong to the level of the maximum membership degree of all factors. Therefore, demand grading results based on urgency are as follows: food, bottled water and medical drugs are extra urgency materials; quilts and tents are urgency materials; shovel is general material, see Table 5.

**Table 4.** Membership degree of all factors under each demand urgency of six emergency materials.

| Emergency Material | Extra Urgent | Urgent | General |
|---|---|---|---|
| Food | 0.36 | 0.10 | 0 |
| Bottled Water | 0.35 | 0.09 | 0 |
| Medical Drugs | 0.35 | 0.08 | 0 |
| Quilt | 0.08 | 0.54 | 0.10 |
| Tent | 0.14 | 0.27 | 0.08 |
| Shovel | 0 | 0.08 | 0.64 |

**Table 5.** Material Grading Results.

| Demand Urgency | Emergency Material |
|---|---|
| Extra Urgent | Food, Bottled Water, Medical drugs |
| Urgent | Quilt, Tent |
| General | Shovel |

The demand quantity of demand points for different grades of materials can be calculated by Formulas (4) and (5), as shown in Table 6.

**Table 6.** Demand of emergency materials at different urgency levels.

| No. | Demand Quantity (Ten Thousand) | | | No. | Demand Quantity (Ten Thousand) | | |
|---|---|---|---|---|---|---|---|
| | Extra Urgent | Urgent | General | | Extra Urgent | Urgent | General |
| 1 | 164.07 | 39.07 | 1.56 | 10 | 28.52 | 6.79 | 0.27 |
| 2 | 78.75 | 18.75 | 0.75 | 11 | 78.33 | 18.65 | 0.75 |
| 3 | 98.11 | 23.36 | 0.93 | 12 | 12.60 | 3.00 | 0.12 |
| 4 | 81.48 | 19.40 | 0.78 | 13 | 25.10 | 5.98 | 0.24 |
| 5 | 140.36 | 33.42 | 1.34 | 14 | 11.26 | 2.68 | 0.11 |
| 6 | 44.69 | 10.64 | 0.43 | 15 | 9.64 | 2.30 | 0.09 |
| 7 | 17.12 | 4.08 | 0.16 | 16 | 138.12 | 32.89 | 1.32 |
| 8 | 15.35 | 3.66 | 0.15 | 17 | 21.53 | 5.13 | 0.21 |
| 9 | 11.07 | 2.64 | 0.11 | 18 | 11.59 | 2.76 | 0.11 |

(2) Reserve siting results

Using heuristic multi center clustering location algorithm in 2.2.3, we can get the results from Formulas (6)–(16), which requires at least 12 reserve points to meet the rescue needs of 18 districts and counties. The 12 reserve points are respectively set in position 1, 2, 5, 6, 7, 8, 9, 10, 11, 12, 13, and 16, and the maximum value of the total effectiveness of rescue is 39.70, including the time effective value of extra urgency materials is 17.79, the time effective value of urgency materials is 17.99, and the time effective value of general materials is 3.92. The average response time of emergency rescue was 0.31 days. The selected location of materials and reserve quantity of graded materials are shown in Table 7.

**Table 7.** Results of the location selection and reserve quantity of graded materials.

| Location | Service Area | Reserve Quantity (Ten Thousand) | | |
|---|---|---|---|---|
| | | Extra Urgent | Urgent | General |
| 1 | 1 | 164.07 | 39.07 | 1.56 |
| 2 | 2, 3, 4 | 258.34 | 61.51 | 2.46 |
| 5 | 5 | 140.36 | 33.42 | 1.34 |
| 6 | 6, 15 | 54.33 | 12.94 | 0.52 |
| 7 | 7 | 17.12 | 4.08 | 0.16 |
| 8 | 8, 14 | 26.61 | 6.34 | 0.26 |
| 9 | 9 | 11.07 | 2.64 | 0.11 |
| 10 | 10, 18 | 40.11 | 9.55 | 0.38 |
| 11 | 11 | 78.33 | 18.65 | 0.75 |
| 12 | 12 | 12.60 | 3.00 | 0.12 |
| 13 | 13, 17 | 46.63 | 11.11 | 0.45 |
| 16 | 16 | 138.12 | 32.89 | 1.32 |

*4.3. Discuss the Influence of Demand Urgency on Location Model*

The simulation results show that the model is feasible and effective. In this section, the model parameters are modified to solve the reserve location layout model with only one-grade material: extra urgent, urgent, or general. Taking the graduation results of emergency supplies as the influencing factors, this paper discusses the influence of demand urgency on location model.

When the value of rescue effectiveness is roughly the same, there are significant differences in location selection results and average response time under different demand levels, as shown in Table 8. Under the extra urgent level, the maximum number of reserve points to be deployed is 4, followed by 3 for urgent type and 2 for general type. The results show that in the case of similar overall rescue satisfaction, the higher the demand urgency, the more investment in infrastructure construction. The shortest response time is 1.84 days for extra urgent materials, 2.73 days for urgent materials and 3.24 days for general materials. According to the phenomenon of similar rescue satisfaction and large difference in response time, it shows that for the materials with higher demand level, the victims hope to get them faster; for the materials with lower demand level, they have higher tolerance and can be delivered later.

**Table 8.** Reserve siting results of materials under different demand urgency.

| Demand Urgency | Number of Reserve Points | Location | Rescue Effectiveness | Average Response Time (Days) |
|---|---|---|---|---|
| Extra urgent | 4 | 4, 7, 8, 13 | 11.36 | 1.84 |
| Urgent | 3 | 7, 9, 12 | 12.33 | 2.73 |
| General | 2 | 1, 8 | 11.16 | 3.24 |

When the number of reserve points is the same, the value of rescue effectiveness is significantly improved after considering the demand graduation. As shown in Table 9, when the number of reserve points is 12, the location results of emergency materials with and without demand graduation are significantly different. In the case of considering the demand graduation, the extra urgent materials are placed in 12 reserves, urgent materials are placed in 9 reserves and general materials are placed in 4 reserves. The total effectiveness value of emergency rescue is 47.42, which is higher than 39.70 when the demand graduation is not considered. The results indicate that: considering the urgency of the demand and reasonably planning the reserve location of different grades of materials, the time effect value of material rescue can be effectively improved.

**Table 9.** Comparison of reserve siting results of materials with and without demand graduation.

| Demand Graduation | Number of Reserve Points | Material Grade | Location | Rescue Effectiveness |
|---|---|---|---|---|
| Yes | 12 | Extra urgent<br>Urgent<br>General | 1, 2, 5, 6, 7, 8, 9, 11, 12, 13, 17, 18<br>1, 5, 6, 7, 8, 9, 12, 17, 18<br>1, 6, 7, 12 | 47.42 |
| No | 12 | No | 1, 2, 5, 6, 7, 8, 9, 10, 11, 12, 13, 16 | 39.70 |

*4.4. Sensitivity Analysis*

In order to show the sensitivity of the model to the parameter Rmax. 50 numerical examples are run by changing the value of Rmax (from 150 km to 400 km, increasing by 5 km) to explore the relationship between Rmax and the number of reserve points, rescue satisfaction and average response time. The operation results are shown in Table 10.

**Table 10.** Number of reserve points, rescue effectiveness and average response time under different Rmax value.

| Rmax (km) | Number of Reserve Points | Rescue Effectiveness | Average Response Time (Days) |
|---|---|---|---|
| 150 | 10 | 40.05 | 0.76 |
| 155 | 9 | 40.13 | 0.94 |
| 160, 165 | 8 | 40.14 | 1.08 |
| 170, 175 | 8 | 39.99 | 1.06 |
| 180, 185, . . . , 195 | 7 | 39.92 | 1.05 |
| 200, 205 | 6 | 39.57 | 1.27 |
| 210, 215 | 5 | 38.26 | 1.78 |
| 220, 225, . . . , 345 | 4 | 37.27 | 2.11 |
| 350, 355, . . . , 385 | 3 | 36.05 | 2.37 |
| 390, 395 | 3 | 31.56 | 2.73 |
| 400 | 3 | 30.77 | 3.03 |

In general, with the increase in Rmax, the number of reserve points needed to be deployed decreased in a ladder like manner, the rescue satisfaction decreased, and the average response time increased. This is the same as what we expected. With the increase in coverage radius, the rescue quality will decrease.

When the Rmax value is 350 to 400 km, the number of reserve points needed to deployed is unchanged at 3, but the rescue effective value decreased by 5.28 and average response time increased by 0.66 day.

Rescue effectiveness and average response time can reflect the rescue service level that the reserve siting plan can provide, while number of reserve points reflects the expenditure that government need to pay. According to the output of the model, the government can weigh the cost and rescue quality, and choose the appropriate location and layout plan of emergency supplies reserve.

**5. Conclusions**

There is a large demand for all kinds of emergency materials when earthquake disaster occurs. In the pre-disaster period, scientifically ranking the priority of various emergency materials and locating them respectively are the guarantee of rapid, accurate rescue.

In this paper, a prepositioning model of emergency materials reserve based on demand graduation is established to solve the problems of demand graduation of multi category materials and the location and layout of various levels of materials reserves before disaster.

First of all, considering the time utility value of materials, emergency materials are graded in accordance with demand urgency. Then, five main factors that influence the demand urgency are put forward, including the degree of life damage caused by insufficient supply, the degree of shortage of emergency materials, the degree of irreplaceable, demand periodicity and demand shelf life. A technical method to determine the demand urgency grade based on fuzzy comprehensive evaluation is developed which can divide the material into three levels: extra urgent, urgent, and general. Then, on the basis of

demand graduation, the "time effective value function" is introduced to represent the satisfaction level of demand under different urgency grade. A location layout model with the dual objectives of "minimum number of deployed reserve points" and "maximum total timeliness value of emergency rescue" is constructed, and a heuristic multi center clustering location algorithm solving model is designed.

Taking the demand graduation as the influencing factor, this paper explores the influence of demand graduation on location selection and layout. The results show that: when the overall rescue satisfaction level is similar, the higher the demand level, the more reserve points need to be deployed. When the same number of reserve points are deployed, the rescue satisfaction after considering the demand graduation is significantly improved. The results show that the emergency material reserve location planning can effectively improve the rescue satisfaction and save financial expenditure by considering the urgency of demand before the disaster. The model was applied to the prepositioning planning of 6 kinds of emergency materials in an earthquake affected area in Sichuan, China and can be used in the reserve siting planning of emergency supplies in any earthquake prone area. According to the output of the model, the government can weigh the cost and rescue quality, and choose the appropriate location and layout plan of emergency supplies reserve.

In the future research, we will further consider multiple modes of transportation decision-making. It is important to determine the fleet size suitable for different transportation requirements before the disaster, because planning in advance and signing an agreement with the transportation company can reduce the risk of insufficient vehicles after the disaster and obtain preferential prices.

**Author Contributions:** Conceptualization, X.Y.; Data curation, X.Y.; Formal analysis, X.Y.; Funding acquisition, H.H.; Methodology, X.Y.; Project administration, H.H.; Software, X.Y.; Writing—original draft, X.Y.; Writing—review and editing, X.Y., H.H., J.Y. and J.F. All authors have read and agreed to the published version of the manuscript.

**Funding:** This research was funded by National Key R&D Plan of China, grant number No. 2016YFC0803207.

**Institutional Review Board Statement:** Not applicable.

**Informed Consent Statement:** Not applicable.

**Data Availability Statement:** Not applicable.

**Conflicts of Interest:** The authors declare no conflict of interest.

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
