# Peer review of "Site Selection and Layout of Material Reserve Based on Emergency Demand Graduation under Large-Scale Earthquake"

_sustainability, doi:10.3390/su13031236_

Round 1

Reviewer 1 Report

The aim of the paper is not clearly defined. In the Introduction chapter, a problem is defined from line 55 to 65. However, the aim of the research investigation is not clearly stated.

The use of methods is well described. 

Numerical results are derived correctly, but they are rather standard.

I believe that hypotheses must be formulated differently. I lack a clear and summary answer to the established hypotheses, confirmation or refutation of assumptions.

A hypothesis (assumption) means a statement whose validity is only assumed, but is also formulated in such a way that it can be confirmed or refuted. When creating correctly formulated hypotheses, care must be taken to observe the three basic rules for the formulation of hypotheses:
1. Hypotheses are claims that need to be formulated as notification sentences and not confused with a research question (problem).
2. Hypotheses express the relationship of at least two variables. This relationship between the two phenomena must be clearly and explicitly expressed. It is appropriate to compare and verify the variables:
differences (more, more often, stronger, higher, different),
relationships (positive, negative association, correlation)
or consequences (by if – then, how – and when – then..).
3. The hypothesis must be testable, it must be confirmed or refuted. Variables must be measured or categorized.

Scientific hypotheses can only be formulated for relational and causal research problems, because the hypothesis is defined as a statement of the relationship between 2 variables.

Authors can change the formulation of hypotheses. Or clearly present your goal and the research question and the hypotheses not to state.

Reviewer 2 Report

Dear author(s),

Please find below the Review Report.

Author Response

Please see the attachment. Thanks~

Round 2

Reviewer 2 Report

The revised version of the manuscript entitled "Site selection and layout of material reserve based on emergency demand graduation under large-scale earthquake" (Manuscript ID: sustainability-1066246) improved in a positive way. The author(s) incorporated in a serious manner the suggested changes and recommendations and pointed out properly all the revisions. Therefore, the manuscript deserves to be published in current form.